# OctreeOcc: Efficient and Multi-Granularity Occupancy Prediction Using Octree Queries

**Yuhang Lu**
ShanghaiTech University
luyh2@shanghaitech.edu.cn

**Xinge Zhu**
The Chinese University of Hong Kong
zhuxinge123@gmail.com

**Tai Wang**[*]
Shanghai AI Laboratory
taiwang.me@gmail.com

**Yuexin Ma**[*]
ShanghaiTech University
mayuexin@shanghaitech.edu.cn

## Abstract

Occupancy prediction has increasingly garnered attention in recent years for its fine-grained understanding of 3D scenes. Traditional approaches typically rely on dense, regular grid representations, which often leads to excessive computational demands and a loss of spatial details for small objects. This paper introduces OctreeOcc, an innovative 3D occupancy prediction framework that leverages the octree representation to adaptively capture valuable information in 3D, offering variable granularity to accommodate object shapes and semantic regions of varying sizes and complexities. In particular, we incorporate image semantic information to improve the accuracy of initial octree structures and design an effective rectification mechanism to refine the octree structure iteratively. Our extensive evaluations show that OctreeOcc not only surpasses state-of-the-art methods in occupancy prediction, but also achieves a $15\% - 24\%$ reduction in computational overhead compared to dense-grid-based methods.

## 1 Introduction

Holistic 3D scene understanding is a pivotal aspect of a stable and reliable visual perception system, especially for real-world applications such as autonomous driving. Occupancy, as a classical representation, has been renascent recently with more datasets support and exploration in learning-based approaches. Such occupancy prediction tasks aim at partitioning the 3D scene into grid cells and predicting semantic labels for each voxel. It is particularly an essential solution for recognizing irregularly shaped objects and also enables the open-set understanding (1), further benefiting downstream tasks, like prediction and planning.

Existing occupancy prediction methods (2; 3; 4; 5; 6) typically construct dense and regular grid representations, same as the ground truth. While such approach is intuitive and direct, it overlooks the statistical and geometric properties of 3D environments. In fact, the 3D scene is composed of foreground objects and background regions with various shapes and sizes. For example, the space occupied by larger objects, such as buses, are considerably more extensive than that taken up by smaller items like traffic cones (Fig. 1a). Consequently, employing a uniform voxel resolution to depict the scene proves to be inefficient, leading to computational waste for larger objects and a lack of geometry details for smaller ones. Considering the large computation cost of aforementioned

---

[*]Corresponding authors. This work was supported by NSFC (No.62206173), Shanghai Frontiers Science Center of Human-centered Artificial Intelligence (ShangHAI), MoE Key Laboratory of Intelligent Perception and Human-Machine Collaboration (KLIP-HuMaCo), ShanghaiTech University.

38th Conference on Neural Information Processing Systems (NeurIPS 2024).

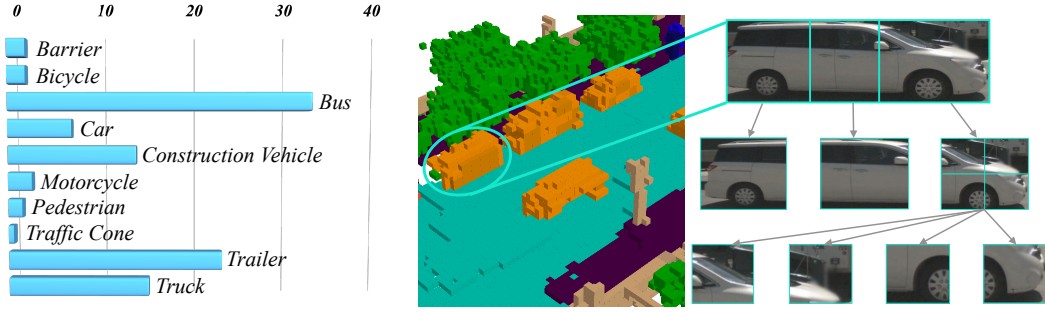

(a) Instance Size Inbalance  (b) 3D and 2D Visualization of Octree Structures

Figure 1: **Scale difference of various categories and octree representation.** (a) compares the average space occupied by different object types, indicating varying granularities needed for different semantic regions. (b) demonstrates the advantage of octree representations, enabling specific granularities for different objects and even parts of objects, reducing computational overhead while retaining spatial information.

works, some recent works attempted to mitigate the heavy memory footprint by utilizing other representations, such as 2D planes in TPVFormer (7) and coarse voxels in PanoOcc (8), or modeling non-empty regions by depth estimation (9). However, these methods suffer from the loss of spatial information because of too coarse representations or accumulated estimated depth errors.

To reduce the computational overhead and meanwhile improve the prediction accuracy, we propose to use octree (10) representation for the occupancy prediction, which can flexibly adapt to objects and semantic regions with various shapes and sizes. As a tree-based data structure, it recursively divides the 3D space into eight octants, thus allowing coarse spatial partition for large regions and fine-grained processing for small objects or complex details (Fig. 1b). Incorporating octree representation, we propose *OctreeOcc*, an efficient and multi-granularity method for occupancy prediction. It constructs octree queries by predicting the octree structure from the stored features of leaf nodes at each level of the tree. However, directly predicting 3D structure from 2D images is challenging due to the lack of depth and occlusion issues. To address this problem, we first propose **Semantic-guided Octree Initialization** that incorporates image semantic information to produce more accurate initial structures. And then, we devise an **Iterative Structure Rectification** module that predicts new octree structure from the encoded query to rectify the low-confidence region of the original prediction, further improving the prediction precision.

Our extensive evaluations against state-of-the-art occupancy prediction methods show that *OctreeOcc* outperforms others on nuScenes and SemanticKITTI datasets, reducing computational overhead by $15\% - 24\%$ for dense-grid-based methods. Ablation studies further validate the effectiveness of each module within our method. Our contributions can be summarized as follows:

- We introduce *OctreeOcc*, a 3D occupancy prediction approach based on multi-granularity octree queries. This method facilitates spatial sparsification, significantly decreasing the number of voxels needed to accurately depict a scene, yet retains critical spatial details.

- We propose a semantic-guided octree initialization module and an iterative structure rectification module, which empower the network with a robust initial setup and the ability to dynamically refine the octree, leading to a more efficient and effective representation.

- Comprehensive experiments demonstrate that OctreeOcc achieves state-of-the-art performance and reduces computational overhead, highlighting the feasibility and potential of octree structures in 3D occupancy prediction.

## 2 Related Work

### 2.1 Camera-based 3D Perception

Camera-based 3D perception has gained significant traction in recent years due to its ease of deployment, cost-effectiveness, and the preservation of intricate visual attributes. According to the view

transformation paradigm, these methods can be categorized into three distinct types. LSS-based approaches(11; 2; 12; 13; 14; 15) explicitly lift multi-view image features into 3D space through depth prediction. Another category of works(16; 17; 18) implicitly derives depth information by querying from 3D to 2D. Notably, projection-free methods(19; 20; 21; 22) have recently demonstrated exceptional performance. While commendable progress has been made in detection, this approach compromises the comprehensive representation of the overall scene in 3D space and proves less effective in recognizing irregularly shaped objects. Consequently, there is a burgeoning interest in methodologies aimed at acquiring a dense voxel representation through the camera, facilitating a more comprehensive understanding of 3D space.

## 2.2 3D Occupancy Prediction

3D occupancy prediction involves the prediction of both occupancy and semantic attributes for all voxels encompassed within a three-dimensional scene, particularly valuable for autonomous vehicular navigation. Recently, some valuable datasets (23; 24; 25) have been proposed, boosting more and more research works (26; 1; 27; 28; 6; 4; 5; 7; 8; 9; 29; 30; 31; 32; 33; 34) in this field. Most of the research focuses on dense voxel modeling. MonoScene(27) pioneers a camera-based approach using a 3D UNet architecture. OccDepth(28) improves 2D-to-3D geometric projection using stereo-depth information. OccFormer(6) decomposes the 3D processing into the local and global transformer pathways along the horizontal plane. SurroundOcc(4) achieves fine-grained results with multiscale supervision. Symphonies(5) introduces instance queries to enhance scene representation.

Nevertheless, owing to the high resolution of regular voxel representation and sparse context distribution in 3D scenes, these methods encounter substantial computational overhead and efficiency issues. Some approaches recognize this problem and attempt to address it by reducing the number of modeled voxels. For instance, TPVFormer(7) models the three-view 2D planes and subsequently recovering 3D spatial information from them. However, its performance degrades due to the lack of 3D information. PanoOcc(8) initially represents scenes at the coarse-grained level and then upsamples them to the fine-grained level, but the lack of information from coarse-grained modeling cannot be adequately addressed by the up-sampling process. VoxFormer(9) mitigates computational complexity by initially identifying non-empty regions through depth estimation and modeling only those specific areas. However, the effectiveness of this process is heavily contingent on the accuracy of depth estimation. In contrast, our approach provides different granularity of modeling for different regions by predicting the octree structure, which reduces the number of voxels to be modeled while preserving the spatial information, thereby reducing the computational overhead and maintaining the accuracy.

## 2.3 Octree-Based 3D Representation

The octree structure(10) finds widespread use in computer graphics for rendering or reconstruction(35; 36; 37; 38), owing to its spatial efficiency and GPU compatibility. Researchers have extended its utility to efficient point cloud learning and related tasks(39; 40; 41; 42). OctFormer(43) and OcTr (44) utilize multi-granularity features of octree to capture a comprehensive global context, thereby enhancing the efficiency of understanding point clouds at the scene level. Furthermore, certain studies (45; 46) adopt octree representation for effectively compressing point cloud data. These works have highlighted the versatility and effectiveness of octree-based methodologies in point cloud analysis and processing applications. However, unlike constructing an octree from 3D point clouds, we are the first to predict the octree structure of a 3D scene from images, which is more challenging owing to the absence of explicit spatial information inherent in 2D images.

## 3 Methodology

In this section, we introduce details of our efficient and multi-granularity occupancy prediction method *OctreeOcc*. After defining the problem and providing an overview of our method in Sec. 3.1 and 3.2, we introduce key components of our method in order. In Sec. 3.3, we outline how we define octree queries and transform dense queries into octree queries. Next, we utilize image semantic priors to construct a high-quality initialized octree structure, as detailed in Sec. 3.4. Once the initialized octree query is obtained, we encode it and refine the octree structure in Sec. 3.5. Finally, Sec. 3.6 describe how to supervise the network.

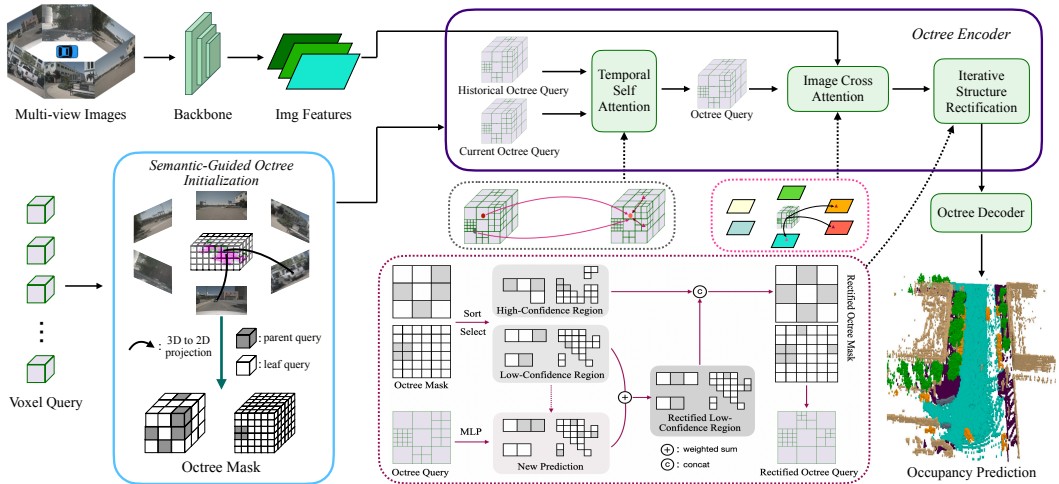

Figure 2: **Overall framework of OctreeOcc.** From multi-view images, we extract multi-scale features using an image backbone. The initial octree structure is derived from image segmentation priors, transforming dense queries into octree queries. The octree encoder refines these queries and rectifies the octree structure. Finally, we decode the octree queries to obtain occupancy predictions. The diagram of the Iterative Structure Rectification module shows the octree query and mask in 2D (quadtree) form for better visualization.

## 3.1 Problem Setup

Camera-based occupancy prediction aims to predict the present occupancy state and semantics of each voxel grid within the scene using input from multi-view camera images. Specifically, we consider a set of $N$ multi-view images $I = \{I_i \in \mathbb{R}^{H \times W \times 3}\}_{i=1}^{N}$, together with camera intrinsics $K = \{K_i \in \mathbb{R}^{3 \times 3}\}_{i=1}^{N}$ and extrinsics $T = \{T_i \in \mathbb{R}^{4 \times 4}\}_{i=1}^{N}$ as input, and the objective of the model is to predict the 3D semantic voxel volume $O \in \{w_0, w_1, ..., w_C\}^{X \times Y \times Z}$, where $H$, $W$ indicate the resolution of input image and $X$, $Y$, $Z$ denote the volume resolution (e.g. $200 \times 200 \times 16$). The primary focus lies in accurately distinguishing the empty class ($w_0$) and other semantic classes ($w_1 \sim w_C$) for every position in the 3D space, which entails the network learning both the geometric and semantic information inherent in the data.

## 3.2 Overview

Given a set of multi-view images $I = \{I_i \in \mathbb{R}^{H \times W \times 3}\}_{i=1}^{N}$, we extract multi-view image features $\mathbb{F} = \{F_i \in \mathbb{R}^{H \times W \times C}\}_{i=1}^{N}$. Simultaneously, we randomly initialize the dense voxel query $Q_{dense} \in \mathbb{R}^{X \times Y \times Z \times C}$. To enhance computational efficiency, we transform $Q_{dense}$ into sparse representation $Q_{octree} \in \mathbb{R}^{N \times C}$, leveraging octree structure information (*i.e.* octree mask) derived from segmentation priors. During encoding, we utilize $Q_{octree}$ to gather information, including temporal fusion and sampling from image features $\mathbb{F}$, while also rectifying the octree structure. Upon encoding $Q_{octree}$, to conform to the output format, we convert it back to $Q_{dense}$ and apply a Multi-Layer Perceptron (MLP) to obtain the final occupancy prediction $O \in \mathbb{R}^{X \times Y \times Z \times K}$. Here, $H$, $W$ indicate the resolution of input image and $X$, $Y$, $Z$ denote the volume resolution. $N$ means the number of octree query ,$C$ denotes the feature dimension and $K$ indicates the number of classes.

## 3.3 Octree Query

Given that objects within 3D scenes exhibit diverse granularities, employing dense queries (6; 4) overlooks this variation and leads to inefficiency. To address this, we propose sparse and multi-granularity octree queries, leveraging the octree structure. This approach creates adaptable voxel representations tailored to semantic regions at different scales.

**Octree Mask.** To effectively construct the octree query, it's essential to understand its underlying structure. An octree partitions each node into eight child nodes within 3D space, each representing

equal subdivisions of the parent node. This recursive process begins with the initial level and proceeds with gradual splitting. At every level, a voxel query serves either as a leaf query if it remains unsplit or as a parent query if it undergoes division. We obtain this geometric information by maintaining a learnable octree mask, denoting as $M_o = \{M_o^l \in \mathbb{R}^{\frac{X}{2^l},\frac{Y}{2^l},\frac{Z}{2^l}}\}_{l=1}^{L-1}$, where $X, Y, Z$ denote the ground truth's volume resolution. The $L$ denotes the depth of the octree, representing the existence of $L$ different resolutions of queries, with $L-1$ splits being performed from the top to the bottom of the octree. The value in the octree mask represents the probability that a voxel at that level requires a split, which is initialized through segmentation priors (Sec. 3.4), rectified during query encoding (Sec. 3.5), and supervised by octree ground truth (Sec. 3.6).

**Transformation between octree query and dense voxel query.** During the query encoding process, we prioritize efficiency by leveraging octree queries. This involves transforming the initial dense queries $Q_{dense}$ into octree queries $Q_{octree}$ using learned structural information. To determine the octree structure, we need to binarise the learned octree mask. Since most of the voxels in the scene at various resolutions do not necessitate splitting, neural network-based prediction binarization is susceptible to pattern collapse, tending to predict all as non-split, leading to a decrease in performance. To mitigate this issue, we introduce a manually defined query selection ratio, where a subset of voxels with the highest probability of splitting is selected for division through the top-k mechanism.

The transformation from $Q_{dense}$ to $Q_{octree}$ begins at the finest granularity, we downsample $Q_{dense}$ to each level through average pooling and retain queries that do not require splitting (leaf queries) with the assistance of the binarized octree mask. This process iterates until reaching the top of the octree. By retaining all leaf queries, we establish $Q_{octree} \in \mathbb{R}^{N \times C}$, where $N = N_1 + N_2 + \ldots + N_L$ represents the total count of leaf queries, L indicates the depth of octree. Conversely, applying the inverse operation of this process allows the conversion of $Q_{octree}$ back into $Q_{dense}$ for the final output. Further details are provided in Appendix.

### 3.4 Semantic-Guided Octree Initialization

Predicting octree structure from an initialised query via neural network can yield sub-optimal results due to the inherent lack of meaningful information in the query. To overcome this limitation, we employ the semantic priors inherent in images as crucial references. Specifically, we adopt UNet(47) to segment the input multi-view images $I$ and obtain the segment map $I_{seg} = \{I_{seg}^i \in \mathbb{R}^{H \times W}\}_{i=1}^N$. We then generate sampling points $p = \{p_i \in \mathbb{R}^3\}_{i=1}^{X \times Y \times Z}$, with each point corresponding to the center coordinates of dense voxel queries. Subsequently, we project these points onto various image views. The projection from sampling point $p_i = (x_i, y_i, z_i)$ to its corresponding 2D reference point $(u_{ij}, v_{ij})$ on the $j$-th image view is formulated as:

$$\pi_j(p_i) = (u_{ij}, v_{ij}), \tag{1}$$

where $\pi_j(p_i)$ denotes the projection of the $i$-th sampling point at location $p_i$ on the $j$-th camera view. We project the point $p_i$ onto the acquired semantic segmentation map $I_{seg}$ through the described projection process. To ensure the prioritization of crucial areas such as foreground objects and buildings in the initial structure, we adopt an unbalanced weight assignment method. Here, the highest weight is allocated to sampling points projecting onto foreground areas, with decreasing weights assigned to points projecting onto buildings or vegetation, and the lowest weight designated for points projecting onto roads, among others. Subsequently, the voxel's weight is determined as the average of the weights of all its sampling points. During this process, we determine the weights of each voxel at the finest granularity, denoted as $W \in \mathbb{R}^{X \times Y \times Z}$. Subsequently, we employ average pooling to downsample $W$ to each level of the octree, resulting in an initial octree mask $M_o = \{M_o^l \in \mathbb{R}^{\frac{X}{2^l},\frac{Y}{2^l},\frac{Z}{2^l}}\}_{l=1}^{L-1}$. Here, $X$, $Y$, and $Z$ represent the resolution of the ground truth volume, while $L$ denotes the depth of the octree. Further details are provided in the Appendix.

### 3.5 Octree Encoder

Given octree queries $Q_{octree}$ and extracted image features $\mathbb{F}$, the octree encoder updates both the octree query features and the octree structure. Referring to the querying paradigm in dense query-based methods(8; 25), we adopt efficient deformable attention(48) for temporal self-attention(TSA) and image cross-attention(ICA).

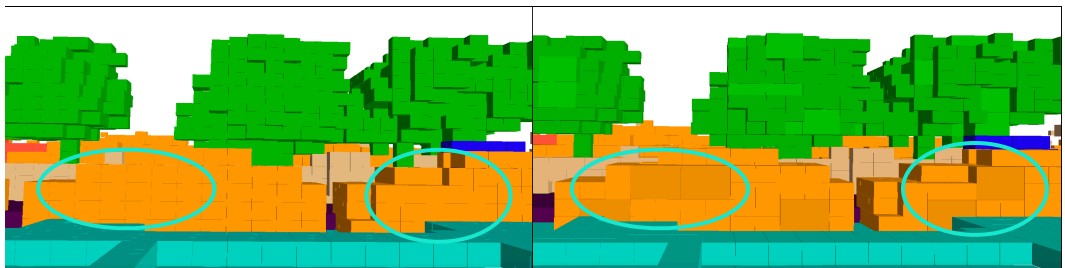

Figure 3: **Illustration of octree structure rectification.** The left figure shows the initially predicted octree structure, while the right figure displays the structure after rectification. It's evident that the rectification module improves the consistency of the octree structure with the object's shape.

In accurately representing the driving scene, temporal information plays a crucial role. By leveraging historical octree queries $Q_{t-1}$, we align it to the current octree queries by the ego-vehicle motion transformation. Given historal octree queries $Q_{t-1} \in \mathbb{R}^{N,C}$, a current octree query $q$ located at $p = (x, y, z)$, the TSA is represented by:

$$TSA(q, Q_{t-1}) = \sum_{m=1}^{M_1} DeformAttn(q, p, Q_{t-1}), \tag{2}$$

where $M_1$ indicates the number of sampling points. Implementing it within the voxel-based self-attention ensures that each octree query interacts exclusively with local voxels of interest, keeping the computational cost manageable.

Image cross-attention is devised to enhance the interaction between multi-scale image features and octree queries. Specifically, for an octree query $q$, we can obtain its centre's 3D coordinate $(x, y, z)$ as reference point $Ref_{x,y,z}$. Then we project the 3D point to images like formula 1 and perform deformable attention:

$$ICA(q, \mathbf{F}) = \frac{1}{N} \sum_{n \in N} \sum_{m=1}^{M_2} DeformAttn(q, \pi_n(Ref_{x,y,z}), \mathbf{F}_n), \tag{3}$$

where $N$ denotes the camera view, $m$ indexes the reference points , and $M_2$ is the total number of sampling points for each query. $\mathbf{F}_n$ is the image features of the $n$-th camera view.

**Iterative Structure Rectification Module.** The initial octree structure, derived from image segmentation priors, may not precisely match the scene due to the potential segmentation and projection errors. Nonetheless, the encoded octree query captures crucial spatial information. Thus, the predicted octree structure based on this query complements and rectifies the initial structure prediction, allowing us to mitigate limitations arising from segmentation and projection issues.

Specifically, we partition the octree structure into two parts: the high-confidence regions and the low-confidence regions, as Fig. 2 shows. By sorting the octree split probability values stored in the octree mask in descending order and selecting the top k% of regions at each level, we can identify the locations of high-confidence regions. For these regions, predictions are relatively more accurate and no additional adjustments are required in this iteration. For regions where confidence remains low, we first extract the query features corresponding to those areas by utilizing the index of low-confidence regions, denoted as $Q_{lcr} = \{Q_{lcr}^l \in \mathbb{R}^{N_l \times C}\}_{l=1}^{L-1}$, where $N_l$ represents the number of low-confidence queries in level $l$. We then employ a MLP to predict octree split probabilities from $Q_{lcr}$. Subsequently, we apply a weighted sum with the previous split probability predictions of low-confidence regions to obtain rectified predictions. These refined predictions are concatenated with the preserved probability predictions of high-confidence regions, culminating in the generation of the final rectified octree mask. It is worth noting that, due to the iterative nature of structure updates, regions initially considered high confidence may not necessarily remain unchanged, as they might be partitioned into low-confidence regions in the next iteration. More details are shown in Appendix.

### 3.6 Loss Function

To train the model, we use focal loss $L_{focal}$, lovasz-softmax loss $L_{ls}$, dice loss $L_{dice}$, affinity loss $L_{scal}^{geo}$ and $L_{scal}^{sem}$ from MonoScene(27). In addition, we also use focal loss to supervise the octree prediction. The overall loss function $L = L_{focal} + L_{ls} + L_{dice} + L_{scal}^{geo} + L_{scal}^{sem} + L_{octree}$.

## 4 Experiments

In this section, we first introduce the datasets (Sec. 4.1), evaluation metrics (Sec. 4.2), and implementation details (Sec. 4.3). Subsequently, we evaluate our method on 3D occupancy prediction and semantic scene completion tasks (Sec. 4.4) to demonstrate its effectiveness. Additionally, we conduct extensive ablation studies and provide more analysis (Sec. 4.5) of our method.

### 4.1 Datasets

**Occ3D-nuScenes(23)** re-annotates the nuScenes dataset(49) with precise occupancy labels derived from LiDAR scans and human annotations. It includes 700 training instances and 150 validation instances, with occupancy spanning -40m to 40m in X and Y axes, and -1m to 5.4m in the Z-axis. Labels are divided into 17 classes, with each class representing a volumetric space of 0.4 meters in each dimension, plus an 18th "free" category for empty regions. The dataset also provides visibility masks for LiDAR and camera modalities.

**SemanticKITTI(50)** comprises 22 distinct outdoor driving scenarios, with a focus on areas located in the forward trajectory of the vehicle. Each sample in this dataset covers a spatial extent ranging from [0.0m, -25.6m, -2.0m, 51.2m, 25.6m, 4.4m], with a voxel granularity set at [0.2m, 0.2m, 0.2m]. The dataset consists of volumetric representations, specifically in the form of 256×256×32 voxel grids. These grids undergo meticulous annotation with 21 distinct semantic classes. The voxel data is derived through a rigorous post-processing procedure applied to Lidar scans.

### 4.2 Evaluation metrics

Both 3D occupancy prediction and semantic scene completion utilize intersection-over-union (mIoU) over all classes as evaluation metrics, calculated as follows:

$$\text{mIoU} = \frac{1}{C} \sum_{c=1}^{C} \frac{\text{TP}_c}{\text{TP}_c + \text{FP}_c + \text{FN}_c}, \tag{4}$$

where $\text{TP}_c$, $\text{FP}_c$, and $\text{FN}_c$ correspond to the number of true positive, false positive, and false negative predictions for class $c_i$, and C is the number of classes.

### 4.3 Implementation Details

Based on previous research, we set the input image size to 900×1600 and employ ResNet101-DCN(51) as the image backbone. Multi-scale features are extracted from the Feature Pyramid Network(52) with downsampling sizes of 1/8, 1/16, 1/32, and 1/64. The feature dimension $C$ is set to 256. The octree depth is 3, and the initial query resolution is 50×50×4. We choose query selection ratios of 20% and 60% for the two divisions. The octree encoder comprises three layers, each composed of TSA, ICA, and Iterative Structure Rectification (ISR) modules. Both $M_1$ and $M_2$ are set to 4. In TSA, we fuse four temporal frames. In ISR, the top 10% predictions are considered high-confidence in level 1, and 30% in level 2. The loss weights are uniformly set to 1.0. For optimization, we employ Adam(53) optimizer with a learning rate of 2e-4 and weight decay of 0.01. The batch size is 8, and the model is trained for 24 epochs, consuming around 3 days on 8 NVIDIA A100 GPUs.

### 4.4 Results

**3D Occupancy Prediction.** In Tab. 1, we compare our method with other SOTA occupancy prediction methods on Occ3d-nus validation set. The performance of FBOCC(3) relies on open-source code, which we evaluate after ensuring consistency in details (utilizing the same backbone, image resolution,

Table 1: **3D Occupancy prediction performance** on Occ3D-nuScenes dataset. "⋆" denotes training with the camera mask.

| Method | Image Backbone | Image Resolution | Reference | mIoU | others | barrier | bicycle | bus | car | const. veh. | motorcycle | pedestrian | traffic cone | trailer | truck | drive. suf. | other flat | sidewalk | terrain | manmade | vegetation |
|---|---|---|---|---|---|---|---|---|---|---|---|---|---|---|---|---|---|---|---|---|---|
| MonoScene(27) | ResNet101 | - | CVPR'22 | 6.06 | 1.75 | 7.23 | 4.26 | 4.93 | 9.38 | 5.67 | 3.98 | 3.01 | 5.90 | 4.45 | 7.17 | 14.91 | 6.32 | 7.92 | 7.43 | 1.01 | 7.65 |
| BEVDet(2) | ResNet101 | - | arxiv'21 | 11.73 | 2.09 | 15.29 | 0.0 | 4.18 | 12.97 | 1.35 | 0.0 | 0.43 | 0.13 | 6.59 | 6.66 | 52.72 | 19.04 | 26.45 | 21.78 | 14.51 | 15.26 |
| BEVFormer(16) | ResNet101 | 900×1600 | ECCV'22 | 23.67 | 5.03 | 38.79 | 9.98 | 34.41 | 41.09 | 13.24 | 16.50 | 18.15 | 17.83 | 18.66 | 27.70 | 48.95 | 27.73 | 29.08 | 25.38 | 15.41 | 14.46 |
| BEVStereo(12) | ResNet101 | - | AAAI'23 | 24.51 | 5.73 | 38.41 | 7.88 | 38.70 | 41.20 | 17.56 | 17.33 | 14.69 | 10.31 | 16.84 | 29.62 | 54.08 | 28.92 | 32.68 | 26.54 | 18.74 | 17.49 |
| TPVFormer(7) | ResNet101 | 900×1600 | CVPR'23 | 28.34 | 6.67 | 39.20 | 14.24 | 41.54 | 46.98 | 19.21 | 22.64 | 17.87 | 14.54 | 30.20 | 35.51 | 56.18 | 33.65 | 35.69 | 31.61 | 19.97 | 16.12 |
| OccFormer(6) | ResNet101 | 896×1600 | ICCV'23 | 21.93 | 5.94 | 30.29 | 12.32 | 34.40 | 39.17 | 14.44 | 16.45 | 17.22 | 9.27 | 13.90 | 26.36 | 50.99 | 30.96 | 34.66 | 22.73 | 6.76 | 6.97 |
| CTF-Occ(23) | ResNet101 | 900×1600 | NeurIPS'23 | 28.53 | 8.09 | 39.33 | 20.56 | 38.29 | 42.24 | 16.93 | 24.52 | 22.72 | 21.05 | 22.98 | 31.11 | 53.33 | 33.84 | 37.98 | 33.23 | 20.79 | 18.00 |
| RenderOcc(26) | ResNet101 | 512×1408 | ICRA'24 | 26.11 | 4.84 | 31.72 | 10.72 | 27.67 | 26.45 | 13.87 | 18.2 | 17.67 | 17.84 | 21.19 | 23.25 | 63.20 | 36.42 | 46.21 | 44.26 | 19.58 | 20.72 |
| BEVDet4D(55)* | Swin-B | 512×1408 | arxiv'22 | 42.02 | 12.15 | 49.63 | 25.1 | 52.02 | 54.46 | 27.87 | 27.99 | 28.94 | 27.23 | 36.43 | 42.22 | 82.31 | 43.29 | 54.46 | 57.9 | 48.61 | 43.55 |
| PanoOcc(8)* | ResNet101 | 900×1600 | CVPR'24 | 42.13 | 11.67 | 50.48 | 29.64 | 49.44 | 55.52 | 23.29 | 33.26 | 30.55 | 30.99 | 34.43 | 42.57 | 83.31 | 44.23 | 54.40 | 56.04 | 45.94 | 40.40 |
| FB-OCC(3)* | ResNet101 | 640×1600 | ICCV'23 | 43.41 | 12.10 | 50.23 | 32.31 | 48.55 | 52.89 | 31.20 | 31.25 | 30.78 | 32.33 | 37.06 | 40.22 | 83.34 | 49.27 | 57.13 | 59.88 | 47.67 | 41.76 |
| Ours* | ResNet101 | 900×1600 | N/A | 44.02 | 11.96 | 51.70 | 29.93 | 53.52 | 56.77 | 30.83 | 33.17 | 30.65 | 29.99 | 37.76 | 43.87 | 83.17 | 44.52 | 55.45 | 58.86 | 49.52 | 46.33 |

Table 2: **3D Semantic Scene Completion** performance on SemanticKITTI dataset.

| Method | Reference | IoU | mIoU | road | sidewalk | parking | other-ground | building | car | truck | bicycle | motorcycle | other-vehicle | vegetation | trunk | terrain | person | bicyclist | motorcyclist | fence | pole | traf-sign |
|---|---|---|---|---|---|---|---|---|---|---|---|---|---|---|---|---|---|---|---|---|---|---|
| LMSCNet(56) | 3DV'20 | 28.61 | 6.70 | 40.68 | 18.22 | 4.38 | 0.00 | 10.31 | 18.33 | 0.00 | 0.00 | 0.00 | 0.00 | 13.66 | 0.02 | 20.54 | 0.00 | 0.00 | 0.00 | 1.21 | 0.00 | 0.00 |
| AICNet(57) | CVPR'20 | 29.59 | 8.31 | 43.55 | 20.55 | 11.97 | 0.07 | 12.94 | 14.71 | 4.53 | 0.00 | 0.00 | 0.00 | 15.37 | 2.90 | 28.71 | 0.00 | 0.00 | 0.00 | 2.52 | 0.06 | 0.00 |
| 3DSketch(58) | CVPR'20 | 33.30 | 7.50 | 41.32 | 21.63 | 0.00 | 0.00 | 14.81 | 18.59 | 0.00 | 0.00 | 0.00 | 0.00 | 19.09 | 0.00 | 26.40 | 0.00 | 0.00 | 0.00 | 0.73 | 0.00 | 0.00 |
| JS3C-Net(59) | AAAI'21 | 38.98 | 10.31 | 50.49 | 23.74 | 11.94 | 0.07 | 15.03 | 24.65 | 4.41 | 0.00 | 0.00 | 6.15 | 18.11 | 4.33 | 26.86 | 0.67 | 0.27 | 0.00 | 3.94 | 3.77 | 1.45 |
| MonoScene(27) | CVPR'22 | 36.86 | 11.08 | 56.52 | 26.72 | 14.27 | 0.46 | 14.09 | 23.26 | 6.98 | 0.61 | 0.45 | 1.48 | 17.89 | 2.81 | 29.64 | 1.86 | 1.20 | 0.00 | 5.84 | 4.14 | 2.25 |
| TPVFormer(7) | CVPR'23 | 35.61 | 11.36 | 56.50 | 25.87 | 20.60 | 0.85 | 13.88 | 23.81 | 8.08 | 0.36 | 0.05 | 4.35 | 16.92 | 2.26 | 30.38 | 0.51 | 0.89 | 0.00 | 5.94 | 3.14 | 1.52 |
| VoxFormer(9) | CVPR'23 | 44.02 | 12.35 | 54.76 | 26.35 | 15.50 | 0.70 | 17.65 | 25.79 | 5.63 | 0.59 | 0.51 | 3.77 | 24.39 | 5.08 | 29.96 | 1.78 | 3.32 | 0.00 | 7.64 | 7.11 | 4.18 |
| OccFormer(6) | ICCV'23 | 36.50 | 13.46 | 58.84 | 26.88 | 19.61 | 0.31 | 14.40 | 25.09 | 25.53 | 0.81 | 1.19 | 8.52 | 19.63 | 3.93 | 32.63 | 2.78 | 2.82 | 0.00 | 5.61 | 4.26 | 2.86 |
| Symphonies(5) | CVPR'24 | 41.92 | 14.89 | 56.37 | 27.58 | 15.28 | 0.95 | 21.64 | 28.68 | 20.44 | 2.54 | 2.82 | 13.89 | 25.72 | 6.60 | 30.87 | 3.52 | 2.24 | 0.00 | 8.40 | 9.57 | 5.76 |
| Ours | N/A | 44.71 | 13.12 | 55.13 | 26.74 | 18.68 | 0.65 | 18.69 | 28.07 | 16.43 | 0.64 | 0.71 | 6.03 | 25.26 | 4.89 | 32.47 | 2.25 | 2.57 | 0.00 | 4.01 | 3.72 | 2.36 |

and excluding CBGS(54)) for a fair comparison. Performance for other methods are reported in a series of works(26; 8; 23). Our approach demonstrates superior performance on mIoU compared to them, particularly excelling in foreground classes such as barriers, cars, and buses, as well as in scene structure classes like manmade and vegetation. This highlights that processing scenes using multiple granularities aligns better with the scene characteristics, enhancing overall expressiveness.

Moreover, we evaluate the efficiency of our approach by comparing it to alternative methods utilizing diverse query forms, as depicted in Tab 3. The results indicate that our approach not only surpasses these methods in terms of performance but also demonstrates significantly reduced memory usage and lower latency compared to dense queries, approaching the levels observed with 2D queries. Clearly, sparse octree queries effectively mitigate computational overhead while ensuring robust performance. Fig. 4 displays qualitative results obtained by our methods and some other methods, illustrating that our approach comprehensively understands the structure of the scene, showcasing superior performance in scene understanding.

**3D Semantic Scene Completion.** To better evaluate the effectiveness of our approach, we conduct comparative experiments for the Semantic Scene Completion (SSC) task. As demonstrated in Tab 2, we compare our results with those of other SSC methods on the SemanticKITTI validation set. Our model demonstrates a more accurate perception of space due to octree construction and correction, outperforming others in IoU metric for geometry reconstruction. Additionally, for specific semantic classes such as car and vegetation, we achieve superior results, attributed to the enhanced treatment of objects facilitated by multi-granularity modeling. Additionally, Tab 3 also indicates that our method consumes fewer computational resources than other dense query-based methods.

Table 3: **Comparison of query form and efficiency** with SOTA methods on the Occ3D-nuScenes (left table) and SemanticKITTI (right table) datasset.

| Methods | Query Form | mIoU | Latency | Memory |
|---|---|---|---|---|
| BEVFormer(16) | 2D BEV | 23.67 | 302ms | 25100M |
| TPVFormer(7) | 2D tri-plane | 28.34 | 341ms | 29000M |
| PanoOcc(8) | 3D voxel | 42.13 | 502ms | 35000M |
| FBOCC(3) | 3D voxels & 2D BEV | 43.41 | 463ms | 31000M |
| Ours | Octree Query | 44.02 | 386ms | 26500M |

| Methods | Query Form | IoU | mIoU | Latency | Memory |
|---|---|---|---|---|---|
| TPVFormer(7) | 2D tri-plane | 35.61 | 11.36 | 179ms | 23000M |
| OccFormer(6) | 3D voxel | 36.50 | 13.46 | 172ms | 22400M |
| VoxFormer(9) | 3D voxel | 44.02 | 12.35 | 177ms | 23200M |
| Symphonics(5) | 3D voxel | 41.44 | 13.44 | 187ms | 22000M |
| Ours | Octree Query | 44.71 | 13.12 | 162ms | 19000M |

Table 4: **Ablation experiments of Modules** on Occ3d-nuScenes *val* set.

| | Baseline | Octree Query | Sem.Init. | Iter.Rec. | mIoU | Latency | Memory |
|---|---|---|---|---|---|---|---|
| (a) | ✓ | | | | 34.17 | 266 ms | 27200M |
| (b) | ✓ | ✓ | | | 34.91 | 218 ms | 18300M |
| (c) | ✓ | ✓ | ✓ | | 36.63 | 214 ms | 17900M |
| (d) | ✓ | ✓ | | ✓ | 35.88 | 227 ms | 18500M |
| (e) | ✓ | ✓ | ✓ | ✓ | **37.40** | 224 ms | 18500M |

Table 5: **Comparison of octree structure quality** at different stages.

| | Stage | mIoU | |
|---|---|---|---|
| | | level 1 to 2 | level 2 to 3 |
| (a) | Initialized w/o unbalanced assignment | 45.79 | 33.60 |
| | Initialized w. unbalanced assignment | 57.34 | 51.28 |
| (b) | After the 1st Rectification | 60.13 | 53.95 |
| | After the 2nd Rectification | 62.27 | 56.79 |

Table 6: **Ablation for different octree depth** on Occ3d-nuScenes *val* set.

| | Query Form | Octree Depth | Query Resolution | mIoU | Latency | Memory |
|---|---|---|---|---|---|---|
| (a) | 3D voxel | N/A | 50×50×4 | 28.51 | 129ms | 7400M |
| (b) | | | 50×50×16 | 31.67 | 186ms | 11600M |
| (c) | | | 100×100×8 | 32.21 | 204ms | 17400M |
| (d) | | | 100×100×16 | 34.17 | 266ms | 27200M |
| (e) | Octree Query | 2 | 50×50×4/100×100×8 | 32.02 | 182ms | 12400M |
| (f) | | 3 | 50×50×4/100×100×8/100×100×16 | 33.76 | 207ms | 16800M |
| (g) | | 4 | 25×25×2/50×50×4/100×100×8/100×100×16 | 34.88 | 193 ms | 15800M |
| (h) | | 3 | 50×50×4/100×100×8/200×200×16 | **37.40** | 224ms | 18500M |

Table 7: **Ablation for the choice of query selection ratio** on Occ3d-nuScenes *val* set.

| | Selection Ratio | mIoU | Latency | Memory |
|---|---|---|---|---|
| (a) | 10%, 60% | 34.47 | 191ms | 14500M |
| (b) | 15%, 60% | 35.01 | 203ms | 16300M |
| (c) | 25%, 50% | 36.73 | 220ms | 18000M |
| (d) | 25%, 60% | 36.12 | 255ms | 21000M |
| (e) | 20%, 60% | **37.40** | 224ms | 18500M |

## 4.5 Ablation Study and More Analysis

In this subsection, we perform ablation studies and analysis experiments on the Occ3d-nus validation set to assess the effectiveness of our proposed modules. All the experiments are conducted on the NVIDIA A40 GPU with reducing the input image size to 0.3x.

**Effectiveness of Octree Queries.** To validate the superiority of octree queries, we maintained consistent TSA and ICA settings while removing the proposed octree structure initialization and rectification modules. This facilitated a comparison with the baseline employing dense queries of size 100×100×16. As illustrated in Tab. 4 (b), experimental results consistently demonstrate our outperformance, with a notable 0.8 mIoU advantage, despite achieving a memory saving of approximately 9G. This underscores the adeptness of octree prediction in allocating queries with varying granularities to diverse semantic regions. Moreover, the constructed octree queries exhibit adaptability to various object shapes, thereby optimizing the utilization of computational resources.

**Effectiveness of Semantic-guided Structure Initialization module.** To highlight the significance of the initial octree structure, we replaced the Semantic-guided Octree Initialization module with an MLP predicting the octree structure from randomly initialized queries. This results in a 1.7 mIoU performance decrease, highlighting the inaccuracy of structurally predicted information from the initialized query due to the absence of valid information coding. Incorporating semantic priors proves crucial as they enhance the quality of the initialized octree, thereby improving overall model performance. Meanwhile, Tab. 5(a) evaluates the effectiveness of the initial octree structure, which shows that assigning different initial weights to voxels based on their semantic regions improves the octree structure by focusing on the scene's effective areas.

**Effectiveness of Iteritive Structure Rectification module.** We perform an ablation study on the Iterative Structure Rectification module, as shown in Tab. 4(d). The incorporation of this module has led to noticeable improvements in performance. Meanwhile, Tab. 5(b) shows this rectification gradually rectifies areas where structural predictions are incorrect. Consequently, this refinement contributes to the efficiency of octree query expression, positively impacting overall performance.

**Discussion on the depth of octree.** Tab. 6 presents experiments on octree depth variations. (a)-(d) show performance with varying 3D query sizes, while (e)-(h) depict octree query performance with different depths and initial resolutions. Comparing (e) to (c) and (f) to (d) reveals our approach achieves comparable performance to dense queries, significantly reducing resource consumption. (g) shows that the predetermined octree depth should not be excessive. While reducing memory usage, the imperfect predictions of octree splitting result in accumulated errors, leading to performance degradation as the depth increases.

**Discussion on query selection ratio of each level in the octree.** In Tab. 7, we present results for different query ratios at various octree levels. Results shows that inadequate queries result in an imperfect scene representation, especially for detailed regions (a,b vs e). Conversely, excessive queries impact computational efficiency, particularly for coarse-grained regions with empty spaces (c

vs d and c,d vs e). Optimizing query numbers based on scene object granularity distribution ensure effective processing of semantic regions of different sizes.

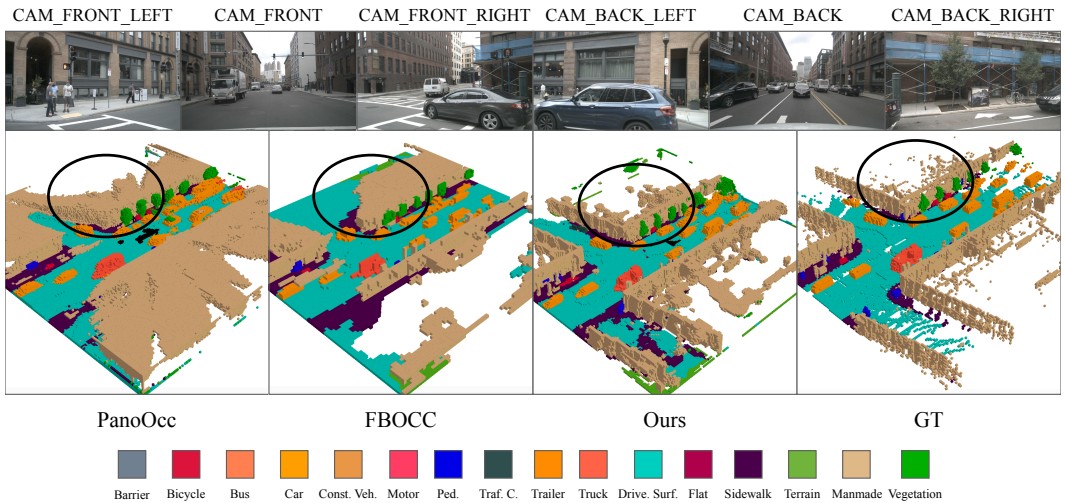

Figure 4: Qualitative results on Occ3D-nuScenes *val* set, where the resolution of the voxel predictions is 200×200×16.

## 5  Conclusions

In conclusion, our paper introduces OctreeOcc, a novel 3D occupancy prediction framework that addresses the limitations of dense-grid representations in understanding 3D scenes. OctreeOcc's adaptive utilization of octree representations enables the capture of valuable information with variable granularity, catering to objects of diverse sizes and complexities. Our extensive experimental results affirm OctreeOcc's capability to attain state-of-the-art performance in 3D occupancy prediction while concurrently reducing computational overhead.

**Limitation.** The quality of the octree ground truth depends on the accuracy of the occupancy ground truth. Current occupancy ground truth comes from sparse lidar point clouds and surface reconstruction, leading to low-quality results for some frames, which affects the octree construction.

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

# Appendix

## A   More Details

In this section, we provide detailed explanations of our proposed modules.

For the **Semantic-Guided Octree Initialization**, our approach commences with acquiring semantic segmentation labels for images by projecting occupancy labels onto the surround-view images. Subsequently, a UNet is trained using these labels. The initialization process entails randomly initializing dense queries, where each query's center point serves as a reference point projected onto the range-view images. If a reference point is projected onto a ground pixel (*i.e.*, driveable surface, other flat, or sidewalk), the probability increases by 0.1. Conversely, if projected onto a background pixel (excluding ground classes), the probability increases by 0.5. Projection onto a foreground pixel increases the probability of requiring a split at that position by 1.0. This process assigns a split probability to each query, and the octree mask is constructed through average pooling, capturing split probabilities at different query levels. After obtaining the octree mask, we designate the top 20% confidence queries as parent queries in level 1, while the remaining queries become Leaf queries and remain unsplit. Moving to level 2, after splitting the parent queries into octants, the top 60% confidence positions are selected as new parent queries, and the remainder as leaf queries. By storing leaf queries at each level, we construct a sparse and multi-granularity octree structure for queries.

In **Iterative Structure Rectification**, at level 1, we retain predictions for the top 10% of positions with confidence. For the remaining positions, a 2-layer MLP is utilized to predict probabilities. These new probabilities are blended with the existing probabilities, with a weight distribution of 60% for the new probabilities and 40% for the old ones. The top 10% of positions with the new probability values are identified as the required splits, shaping the structure of the new level 1. Similarly, at level 2, predictions for the top 30% of positions with confidence are preserved. For positions not in the top 30%, probabilities are predicted using a 2-layer MLP. The new probabilities are computed by merging them with the original probabilities, with an even weight distribution of 50% for each. The top 30% of the new probability values are then selected as the positions necessitating splitting, delineating the structure of the new level 2.

## B   Octree node index calculation

The hierarchical structure of the octree, particularly the assignment of queries to respective levels, is determined based on the octree mask $M_o$ and the query selection ratio denoted as $\alpha = \{\alpha^1, \alpha^2, \ldots, \alpha^{L-1}\}$. These ratios govern the number of subdivisions at each level, thereby defining the hierarchical organization of the octree. The procedure is as follows. For level l, queries with $M_o^l$ values within the top $\alpha^l$ percentile are identified as candidates for octant subdivision. Subsequently, within octants that have undergone one previous subdivision at the next level, queries are once again selected based on their values falling within the top $\alpha^2$ percentile, initiating another round of subdivision. This process continues iteratively until reaching the final level of the octree.

Simultaneously, exploiting the octree structure facilitates the direct conversion of sparse octree queries into dense queries to align with the desired output shape. For a query $q_{octree}$ at level $l$ with the index $(a, b, c)$, the indexes of its corresponding $8^{L-l}$ children nodes in level $L$ are determined by $(a \times 2^{L-l} + a_{offset}, b \times 2^{L-l} + b_{offset}, c \times 2^{L-l} + c_{offset})$, where $a_{offset}$, $b_{offset}$, and $c_{offset}$ are independent, ranging from 0 to $2^{L-l}$. Here, $L$ denotes the depth of the octree. During this process, we allocate the feature of $q_{octree}$ to all of these positions. By iteratively applying this procedure to all queries at each level, we effectively transform $Q_{octree}$ into $Q_{dense}$.

## C   Octree Ground Truth Generation

We derive the octree ground truth from the semantic occupancy ground truth. Specifically, for a voxel at level $l$ in the octree, we identify its corresponding $8^{L-l}$ voxels in the semantic occupancy ground truth. If these voxels share the same labels, we deem the voxel at level $l$ unnecessary to split (assigned a value of 0); otherwise, it necessitates division (assigned a value of 1), as the current resolution is insufficient to represent it adequately. Through this process, each voxel at each level is assigned a

binary value of 0 or 1. Then we obtain the octree ground truth $G_{octree} = \{G_{octree}^l \in \mathbb{R}^{\frac{X}{2^l}, \frac{Y}{2^l}, \frac{Z}{2^l}}\}_{l=1}^{L-1}$. Here, $L$ represents the depth of the octree, while $X$, $Y$, and $Z$ denote the volume resolution of the semantic occupancy ground truth. $G_{octree}$ is employed to supervise the octree mask using focal loss, facilitating the network in learning the octree structure information.

## D    More discussion of octree initialization

Given that the FloSP method outlined in MonoScene(27) incorporates a 3D to 2D projection operation, similar to our initialization approach, we additionally adapted this method for comparison. Specifically, we employed FloSP to extract 3D voxel features from 2D image features. Subsequently, we applied a Multi-Layer Perceptron (MLP) to predict the splitting probability of each voxel, replacing the randomly initialized queries used in the original ablation experiments. The results indicate that, although this operation outperforms predictions from randomly initialized queries, it is still constrained by insufficient information, resulting in a decline in overall performance.

Table 8: More ablation of octree initialization

|      | Initialization Method | mIoU |
|------|-----------------------|------|
| (a)) | Randomly initialised queries | 34.91 |
| (b)  | Voxel features from FLoSP | 35.72 |
| (c)  | Semantic-Guided Octree Initialization | **37.40** |

## E    Analysis of Various Usage of Octree.

As a classic technique, octree is employed in various tasks (42; 35; 36; 37; 38). Despite differences in addressed problems, we compare our method with OGN(42), which proposes an octree-based upsampling approach. We keep the similar setup in Tab. 4 (b) but substitute the deconvolution decoder with OGN's octree decoder. Results in Tab. 9 indicate that employing octree solely in the decoder fails to mitigate excessive computational costs and yields sub-optimal performance, mainly due to the high query count during encoding.

Table 9: Comparison with another octree method.

|          | mIoU | Latency | Memory |
|----------|------|---------|--------|
| baseline | 34.10 | 266 ms | 27200M |
| OGN(42)  | 33.39 | 212 ms | 24300M |
| Ours     | **37.40** | 224 ms | 18500M |

## F    More Visualization

Fig. 5 shows additional visualizations of proposed OctreeOcc. Evidently, our approach, leveraging the multi-granularity octree modeling, demonstrates superior performance particularly in the categories of truck, bus, and manmade objects.

Fig. 6 illustrates the results of occupancy prediction alongside the corresponding octree structure. For clarity in visualization, we employ distinct colors to represent voxels at various levels of the octree prediction, based on their occupancy status. For improved visualization, only a portion correctly corresponding to the occupancy prediction is displayed, rather than the entire octree structure, ensuring clarity and focus on the relevant information. Level 3 (voxel size: 0.4m × 0.4m × 0.4m) is depicted in light gray, level 2 (voxel size: 0.8m × 0.8m × 0.8m) in medium gray, and level 1 (voxel size: 1.6m × 1.6m × 1.6m) in dark gray. It's worth noting that level 1 voxels, predominantly situated in free space and within objects, might be less intuitively discernible. Nonetheless, this image underscores the efficacy of octree modeling, which tailors voxel sizes to different semantic regions, enhancing representation accuracy.

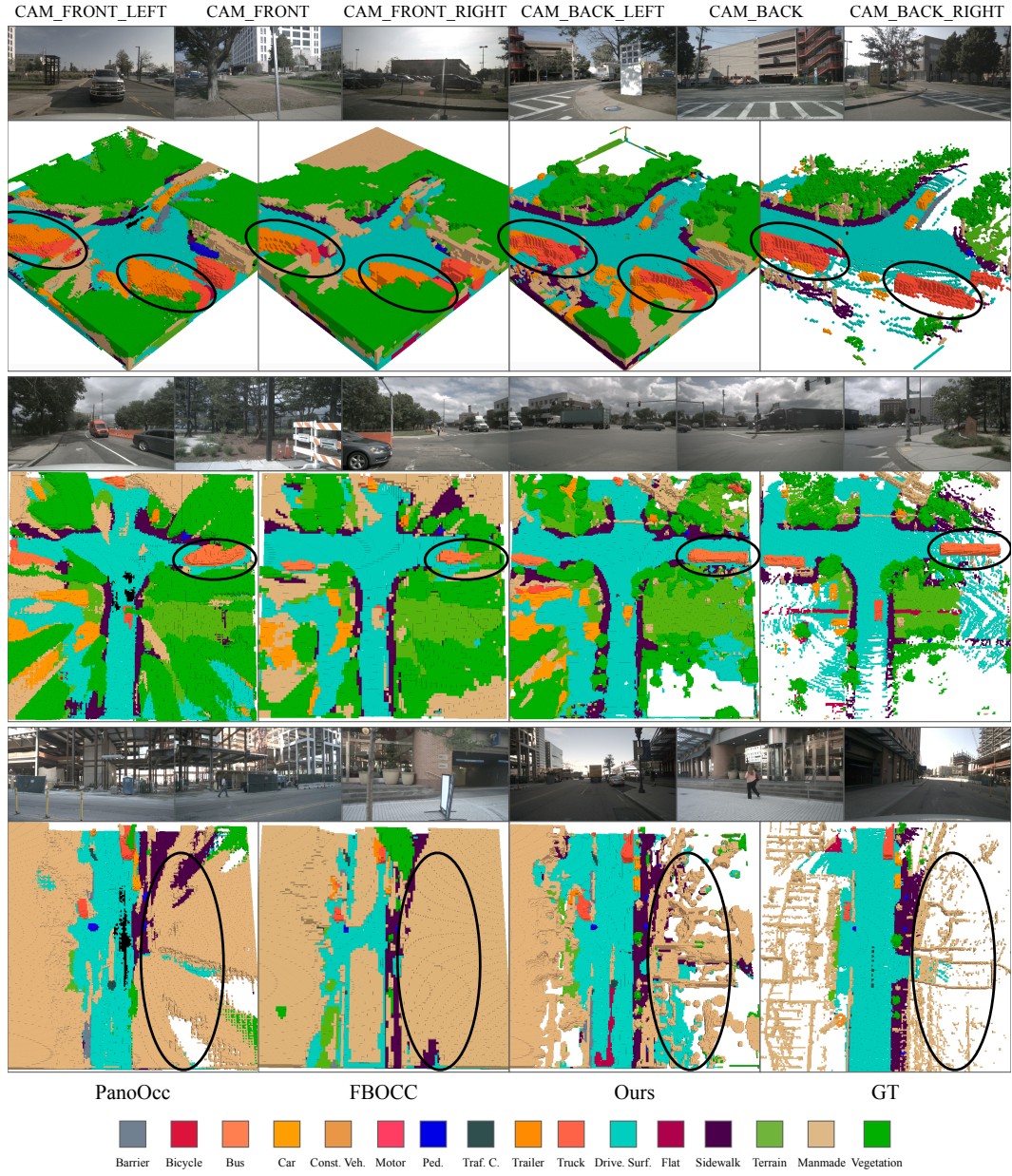

Figure 5: **More visualization on Occ3D-nuScenes validation set.** The first row displays input multi-view images, while the second row showcases the occupancy prediction results of PanoOcc(8), FBOCC(3), our methods, and the ground truth.

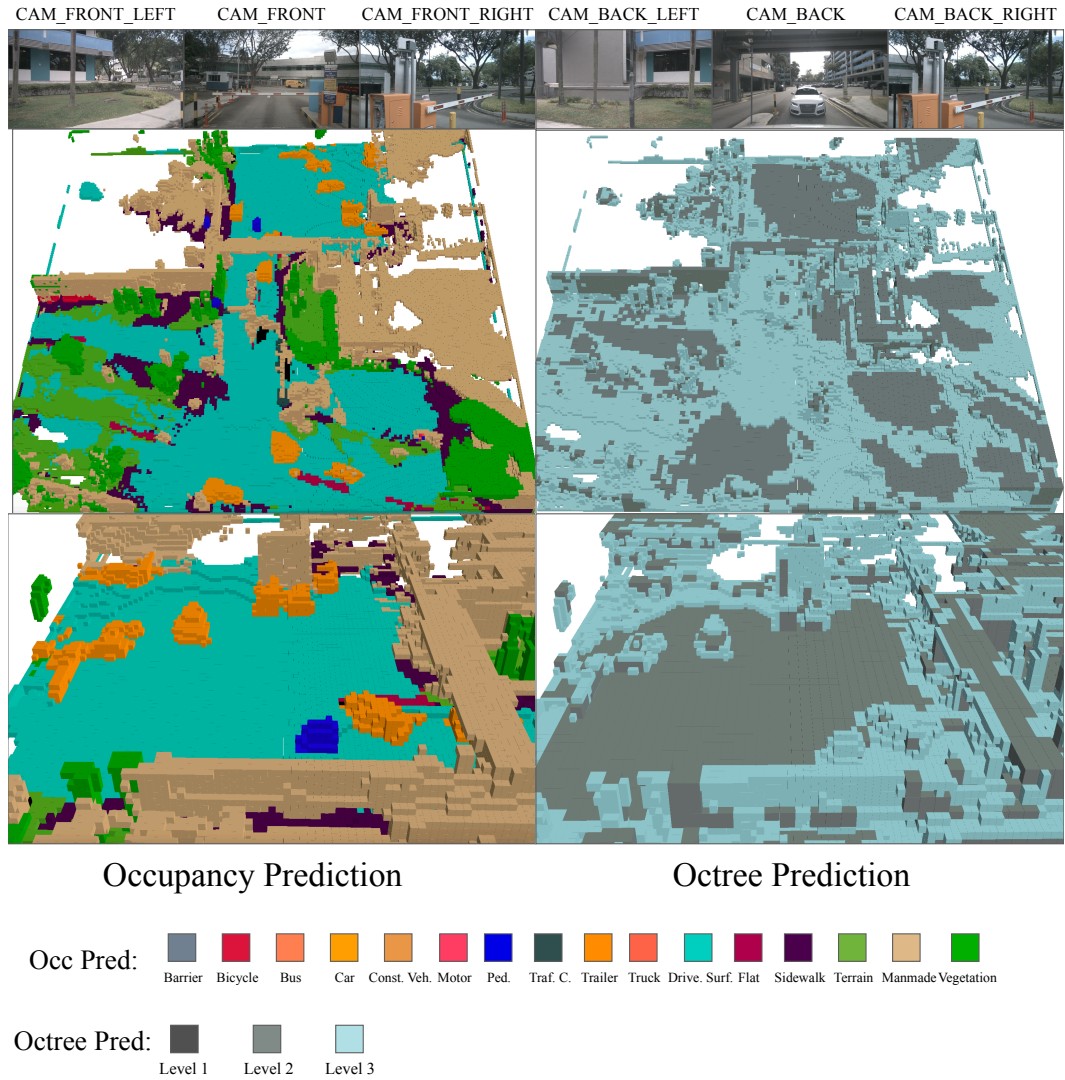

CAM_FRONT_LEFT  CAM_FRONT  CAM_FRONT_RIGHT  CAM_BACK_LEFT  CAM_BACK  CAM_BACK_RIGHT

Occupancy Prediction                    Octree Prediction

Occ Pred: Barrier Bicycle Bus Car Const. Veh. Motor Ped. Traf. C. Trailer Truck Drive. Surf. Flat Sidewalk Terrain Manmade Vegetation

Octree Pred: Level 1 Level 2 Level 3

Figure 6: **Visulization of octree structure.** The first row displays input multi-view images, while the second and third rows showcase the occupancy prediction results and the corresponding octree structure prediction results.

