# OpenReview forum: "OctreeOcc: Efficient and Multi-Granularity Occupancy Prediction Using Octree Queries"
_NeurIPS.cc/2024/Conference — NeurIPS 2024 poster_

### Official Review · Reviewer_mQ4T · 2024-07-03

**Soundness:** 3
**Presentation:** 3
**Contribution:** 2
**Rating:** 5
**Confidence:** 5

**Summary:**

This paper proposes a new occupancy predictor that utilizes the octree structure to reduce the total number of occupancy query. The octree tree is initialized with semantic information and recursively rectified, so that it can accurately represent the structure of the driving scenes.

**Strengths:**

1. This work gives a new way to represent occupancy instead of voxel feature or dense query, which is enlightening for later works.
2. The prediction precision of OctreeOcc exceeds FB-Occ and PanoOcc, indicating a new SOTA. The visualization results also demonstrate its ability to predict fine structures.

**Weaknesses:**

1. OctreeOcc requires extra segmentation model to provide the reference for the initialization of octree queries, which reduces the efficiency gain. More details of the segmentation model should be published.
2. The ablation study shows that the accuracy is largely improved after using the extra semantic information. A fair comparison between OctreeOcc without extra semantic information and other methods is missing.

**Questions:**

1. It is not clear how the high-confidence and low-confidence regions are determined in Iterative Structure Rectification. Is the confidence related to the split probability?
2. How does OctreeOcc handle the conflict between the octree queries initialized by current semantic information and the octree queries aligned from the previous frame?

**Limitations:**

The authors have described the limitation of this work, and there is no potential negative societal impact.

---

> ### Author Rebuttal · Authors · 2024-08-07
>
> W1. **More details of the segmentation model**
>
>    Segmentation models are trained using homologous data and then integrated into the network as fixed components, being inferred together rather than as additional separate modules. In this setup, a UNet architecture is used, where the encoder part is shared with the backbone's ResNet, and the decoder consists of a three-layer deconvolution. This design minimizes additional memory overhead.
>
>    Thank you for the suggestion. We will include these details in the revised version.
>
>
> W2. **Comparison between OctreeOcc without extra semantic information and other methods**
>
>    The **input and output** of our network are **the same** as those of other methods (i.e., input surround-view images and output occupancy prediction result). We trained a sub-network using the same data and incorporated it into the overall framework without relying on additional semantic information. The segmentation network is an integral part of our model, not a separate component. Theoretically, **we do not use extra information**.
>
>    Semantic segmentation is useful for initializing the octree structure; however, even without this component, our network's performance remains comparable to the dense voxel method (as shown in the table below).
>
>    | Method                   | mIoU  | Memory |
>    | ------------------------ | ----- | ------ |
>    | PanoOcc                  | 42.13 | 35000M |
>    | FB-OCC                   | 43.41 | 31000M |
>    | Ours                     | 44.02 | 26500M |
>    | Ours w/o sem-guided init | 42.08 | 25700M |
>
>
> Q1. **How the high-confidence and low-confidence regions are determined**
>
>    As shown in Section 3.5 of original paper, a split probability is maintained throughout the process, dividing high and low confidence areas using this probability and top-k. In this context, confidence actually refers to the split probability.
>
>
> Q2. **How does OctreeOcc handle the conflict between the octree queries initialized by current semantic information and the octree queries aligned from the previous frame?**
>
>    During temporal fusion, the octree query of the historical frame must first be converted into the same structure as the current frame during ego-motion conversion to ensure that fusion can be performed.

---

> > ### Comment · Reviewer_mQ4T · 2024-08-11
> >
> > Thank you for your response. Most of my previous concerns have been resolved. Has the code for this work been open sourced? I believe that open source will have a promoting effect on the development of this field.

---

> > > ### Author Response · Authors · 2024-08-12
> > >
> > > We are pleased to open-source our code to promote the field. Due to the rebuttal rules, we can only send link to the AC at this time. We have sent the code's anonymized link to the AC and promise to make the code public once the paper is accepted.

---

> > > > ### Comment · Reviewer_mQ4T · 2024-08-12
> > > >
> > > > Thank you for your response. My concerns have been addressed, and I am inclined to accept this work. The content of the rebuttal and the code's link should be added to the final version.

---

### Official Review · Reviewer_NcWJ · 2024-07-07

**Soundness:** 3
**Presentation:** 3
**Contribution:** 3
**Rating:** 7
**Confidence:** 4

**Summary:**

This paper introduces OctreeOcc, aiming to tackle the heavy computational demands of the dense and regular grid representations employed by the previous methods. Instead of randomly initializing the octree structure, OctreeOcc incorporates the semantic priors of images as guidance. The octree structure are further updated iteratively to correct the potential errors.

**Strengths:**

1. Octree representation is a good solution to the heavy computation burden of the dense grid representation. OctreeOcc presents the details of the network components clearly.
2. OctreeOcc provides detailed ablation experiments.
3. OctreeOcc achieves good balance between the accuracy and the efficiency.

**Weaknesses:**

1. The image resolution and visible mask should be marked for fairly comparing with other methods. To my understanding, using the visible mask or not results in significant performance differences.
2. This paper mentioned temporal fusion in line 131. Is the temporal information obtained similar to the bevformer? Besides, how many frames are employed? (e.g., 4 frames in the PanoOcc)
3. The metrics of the Symponies in the tables should be updated. The performance differences are acceptable because Symponies uses stereo information (If OctreeOcc were to rely solely on monocular input).
4. Dose the Memory presented in the tables refer to training memory? To my understanding, the inference memory for most of the methods are highly lower than this.
5. In Table 6, query form of octree attend to the image features in a hierarchical manner, what is the number of the attention layers of the 3D voxel?

**Questions:**

Please see above weaknesses.

**Limitations:**

The authors have adequately addressed the limitations.

---

> ### Author Rebuttal · Authors · 2024-08-07
>
> W1. **The image resolution and visible mask should be marked**
>
>    In Table 1 of original paper, we use "$\star$" to indicate whether a method uses a camera mask during training. Methods marked with "$\star$" are the latest SOTA methods.
>
>    The image sizes used in training for each method are as follows:
>
>    | Method    | Image Resolution |
>    | --------- | ---------------- |
>    | BEVFormer | 900 × 1600       |
>    | TPVFormer | 900 × 1600       |
>    | OccFormer | 896 × 1600       |
>    | CTF-Occ   | 900 × 1600       |
>    | RenderOcc | 512 × 1408       |
>    | BEVDet4D  | 512 × 1408       |
>    | PanoOcc   | 900 × 1600       |
>    | FB-OCC    | 640 × 1600       |
>    | Ours      | 900 × 1600       |
>
>    All methods use settings close to full-size images within their respective ranges, ensuring a fair comparison.
>
> W2. **Temporal Fusion Details**
>
>    Yes, the temporal fusion method is the 3D version of BEVFormer’s TSA, using a fusion of four frames, consistent with PanoOcc.
>
> W3. **The metrics of the Symponies in the tables should be updated**
>
>    Thanks for your reminding, we will follow the number in its updated version and update it in our final version.
>
> W4. **Memory presented in the tables**
>
>    Yes, the table shows memory consumption during training.
>
> W5. **The number of the attention layers of the 3D voxel**
>
>    As shown in implementation details, octree encoder consists of three layers, each composed of TSA, ICA, and ISR modules.

---

> > ### Comment · Reviewer_NcWJ · 2024-08-09
> >
> > Thank you for the rebuttal. My concerns have been addressed, and I am inclined to accept this work. The content of the rebuttal should be added to the revised version.

---

> > > ### Author Response · Authors · 2024-08-09
> > >
> > > Thank you for raising your score. We will include the details mentioned in the rebuttal in the revised version.

---

### Official Review · Reviewer_5724 · 2024-07-09

**Soundness:** 3
**Presentation:** 2
**Contribution:** 2
**Rating:** 5
**Confidence:** 5

**Summary:**

This paper introduces OctreeOcc, an innovative 3D occupancy prediction framework that leverages the octree representation to adaptively capture valuable information in 3D, offering variable granularity to accommodate object shapes and semantic regions of varying sizes and complexities. The authors incorporate image semantic information to improve the accuracy of initial octree structures and design an effective rectification mechanism to refine the octree structure iteratively. Extensive evaluations show that OctreeOcc not only surpasses state-of-the-art methods in occupancy prediction, but also achieves a 15% − 24% reduction in computational overhead compared to dense-grid-based methods.

**Strengths:**

- OctreeOcc utilises octree to represent 3D space, which is novel and no one has done it before as far as my knowledge.

- Owing to the design of octree queries, OctreeOcc improves the accuracy of occupancy prediction with the latency also decreasing.

- Extensive experiments on Occ3D and SemanticKitti demonstrate the effectiveness of OctreeOcc.

**Weaknesses:**

- The pipeline is quite complex and heavily relies on manual operations, such as the selection strategy in Iterative Structure Rectification Module. My concern is that the complicated design may reduce the generalization.

- The updating processing of octree queries is not very clear. In my understanding, if the octree mask changes, the newly generated octree queries cannot directly match the original octree queries.

- Different octree queries may have different granularity, but, as shown in Eql (3), all octree queries seem to interact with image features only referring to their centers, which ignoring the influence of granularity.

**Questions:**

please refer to weaknesses.

**Limitations:**

NaN

---

> ### Author Rebuttal · Authors · 2024-08-07
>
> W1. **Too many manually set hyperparameters affect generalisability**
>
>    Octree-related hyperparameters focus on selecting the top-k operation’s K during octree query sparsification and rectification.
>
>    These hyperparameters are statistically derived from the dataset and are not difficult to determine.
>
>    The experiments conducted on two datasets in the original paper used **the same set of octree hyperparameters**. Table 2 shows that applying the octree hyperparameters from nuScenes to SemanticKITTI resulted in SOTA performance on the IoU metrics and performance comparable to SOTA on the mIoU metrics. These experiments validate the generality of our approach.
>
> W2. **Updating processing of octree queries**
>
>    When the octree mask changes, the strategy for adjusting queries in the affected region is to first convert to a dense form and then adapt to the new sparse structure based on the updated octree mask.
>
> W3. **Octree queries focus only on their centers, neglecting granularity effects**
>
>    Thank you for the suggestion. We have compared the sampling scheme you mentioned (as shown in the Table below), where we assign different numbers of reference points to queries of varying granularity during cross-attention. For example, the finest granularity query uses only its centroid, while the medium granularity query samples four additional points around its centroid.
>
>    This approach can lead to performance gains but also increases memory overhead due to the effective increase in the number of queries. Our original design **balances memory overhead with performance**; regions with coarse query granularity are likely part of the same object, so sampling features only at the query's center is adequate. Overall, finding better ways to utilize octree queries will be our next step in enhancing our work.
>
>    | Sampling Method      | mIoU  | Memory |
>    | -------------------- | ----- | ------ |
>    | Original             | 37.40 | 18500M |
>    | More sampling points | 38.12 | 23100M |

---

> > ### Comment · Reviewer_5724 · 2024-08-14
> >
> > Thank the authors for responding my concerns. Most of my concerns are addressed, but I still consider the pipeline is too complicated in updating octree queries.
> > However, it will not affect I think this work is a solid work and I will maintain my positive assessment of this paper.

---

### Official Review · Reviewer_Mixn · 2024-07-10

**Soundness:** 3
**Presentation:** 3
**Contribution:** 3
**Rating:** 6
**Confidence:** 4

**Summary:**

The authors aim to tackle the problem of high memory usage in dense occupancy prediction for 3D scenes. They introduce OctreeOcc, a method that uses octree structures to make predictions more efficiently. Experimental results show that OctreeOcc reduces computational load and achieves competitive performance.

**Strengths:**

S1: The authors address the significant issue of high memory consumption in dense occupancy prediction, highlighting the importance of finding more efficient solutions.

S2: The use of octree representation for occupancy prediction is a novel and effective approach, offering adaptive granularity for various object shapes and regions.

S3: Experimental results show some reduction in memory usage and latency, demonstrating the efficiency of the proposed method.

**Weaknesses:**

W1: The proposed method seems difficult to optimize and relies on segmentation and historical information.

W2: The results on the SemanticKITTI dataset do not achieve state-of-the-art performance, and the reduction in memory usage is not very significant.

W3: The training process is complex and costly, requiring three days on 8 A100 GPUs, which seems less efficient compared to previous methods.

**Questions:**

Q1: The proposed method seems to be quite time-consuming to train. Can the authors provide a comparison of training times with other methods? This is especially relevant given the emphasis on efficiency throughout the paper.

Q2: There is some confusion regarding the reported memory consumption in the experiments. For instance, the original VoxFormer [1] paper reports memory usage of less than 16GB, but the authors here report around 23GB. What accounts for this significant discrepancy? Are there differences in the settings or configurations used?

Q3: The implementation details provided focus on the nuScenes dataset. Could the authors include the specifics of the implementation for the SemanticKITTI dataset as well?

Q4: The paper lacks results or discussions on other commonly used datasets, such as SSCBench-KITTI360 [2]. Including these would provide a more comprehensive evaluation of the proposed method.

[1] VoxFormer: Sparse Voxel Transformer for Camera-based 3D Semantic Scene Completion

[2] SSCBench: Monocular 3D Semantic Scene Completion Benchmark in Street Views

**Limitations:**

The authors have identified a significant limitation in their work: the dependency on the quality of the occupancy ground truth (GT). This reliance can lead to suboptimal performance if the GT data, derived from sparse LiDAR point clouds and surface reconstruction, is not accurate or complete.

---

> ### Author Rebuttal · Authors · 2024-08-07
>
> W1. **The proposed method seems difficult to optimize and relies on segmentation and historical information**
>
>    Regarding optimization, our method uses the same optimizer settings as other methods and is trained for 24 epochs as well. Under **the same training conditions**, our approach achieves SOTA performance on multiple benchmarks.
>
>    Our segmentation component is integrated into the model as a whole and is used to obtain a more accurate initialization. Importantly, **the input to the framework remains unchanged**, continuing to use surround-view images. As shown in Table 4 of the original paper, even without semantic-guided initialization, our approach still outperforms the baseline (34.17 vs. 35.88) while reducing memory consumption (27200M vs. 18500M).
>
> Furthermore, incorporating historical information is **standard** in all occupancy prediction methods, including ours. Including historical information is optional, and using only the current frame does not affect the model's inference.
>
> W2. **Results on the SemanticKITTI**
>
>    In our experiment on SemanticKITTI, we used the same set of octree hyperparameters as for nuScenes, achieving SOTA IoU, which highlights the effectiveness of our approach in capturing the overall spatial structure.
>
> For smaller objects (e.g., bicyclists, poles), our IoU is limited by the three-level octree structure. The 32 × 32 × 4 resolution of the first level may not fully capture small objects, leading to early information loss. This represents a **trade-off** between performance and computational resources; increasing the number of queries generally improves performance but significantly raises memory overhead.
>
> Regarding memory overhead reduction, since model sizes for different methods on SemanticKITTI are small, the potential for memory reduction is limited. However, our memory reduction percentage on SemanticKITTI remains considerable.
>
> Overall, the current results sufficiently validate the effectiveness of our approach. We will continue to fine-tune the hyperparameters for SemanticKITTI and explore better trade-offs to achieve improved performance across different datasets.
>
> W3&Q1. **The training time**
>
>    We present the results of the comparison of the training time of each model on the nuScenes dataset, which show that our method **does not significantly increase training time** compared to other methods.
>
>    | Method    | Training Time |
>    | --------- | ------------- |
>    | BEVFormer | 61h           |
>    | PanoOcc   | 67h           |
>    | FBOCC     | 63h           |
>    | Ours      | 71h           |
>
> Q2. **Memory consumption of VoxFormer**
>
>    The memory consumption reported in the VoxFormer paper accounts only for the stage-2. For a fair comparison, our calculations include both the stage-1 and stage-2. Additionally, the memory consumption we report for our method also includes the segmentation part of the model.
>
> Q3. **Implementation details for SemanticKITTI**
>
>    The setup for SemanticKITTI remains essentially the same as in the previous method. We use ResNet50 as the backbone with an image size of 1220x370. The Adam optimizer is employed, with a learning rate of 2e-4 and a weight decay of 0.01. The octree-related hyperparameters are consistent with those set for the nuScenes dataset. We will include these details in the final version.
>
> Q4. **Results on SSCBench-KITTI360**
>
> We provide experiment results on SSCBench-KITTI360, and with the same efficient framework and settings, our method also outperforms other methods in both metrics.
> | Method    | IoU   | mIoU  |
> | --------- | ----- | ----- |
> | TPVFormer | 40.22 | 13.64 |
> | OccFormer | 40.27 | 13.81 |
> | VoxFormer | 37.76 | 11.91 |
> | Ours      | 40.89 | 14.03 |

---

> > ### Comment · Reviewer_Mixn · 2024-08-13
> >
> > I appreciate your efforts in addressing my concerns in the rebuttal. Based on your responses, I am increasing my score to weak accept.

---

### Decision · Program_Chairs · 2024-09-25

**Decision:**

Accept (poster)

**Comment:**

The manuscript presents a novel efficient method for occupancy prediction in LIDAR scans. The proposed method uses octrees to reduce the computational footprint of previous methods while retaining competitive generalization performance. The reported improvements are quite incremental, but future work may lead to further improvements.